# Impact of pH and protein hydrophobicity on norovirus inactivation by heat-denatured lysozyme

**Michiko Takahashi[1¤], Hajime Takahashi[1]\*, Yumiko Okakura[1], Masahiro Ichikawa[2], Takashi Kuda[1], Bon Kimura[1]**

**1** Department of Food Science and Technology, Faculty of Marine Science, Tokyo University of Marine Science and Technology, Tokyo, Japan, **2** Kewpie Corporation, Tokyo, Japan

¤ Current address: Faculty of Science and Technology, Kochi University, Kochi, Japan.
* hajime@kaiyodai.ac.jp

**Data Availability Statement:** All relevant data are within the manuscript and its Supporting Information files.

**Funding:** This work was supported by Grant-in-Aid for JSPS Fellows Grant Number 17J05482, and

## Abstract

Norovirus, the leading cause of non-bacterial food poisoning, is responsible for several outbreaks associated with bivalves and ready-to-eat food products worldwide. As norovirus is resistant to alcohol, which is commonly used in food manufacturing processes, sodium hypochlorite is used for its inactivation. However, sodium hypochlorite has two disadvantages: it cannot be added to foods, and its effect is significantly reduced in the presence of organic compounds. Thus, a novel disinfectant against norovirus is urgently required for food hygiene. Thermally denatured egg white lysozyme inactivates norovirus; however, the optimal inactivating conditions and the underlying mechanism are unclear. In the present study, the inactivating mechanism of heat-denatured lysozyme against norovirus was analyzed using murine norovirus strain 1 (MNV-1). We found that the inactivating effect was enhanced by adjusting the pH of the lysozyme solution before thermal denaturation to 6.5 or higher. The reaction of heat-denatured lysozyme and MNV-1 was irreversible, and norovirus was completely inactivated after exposure to heat-denatured lysozyme. Furthermore, it was found that lysozyme residues 5–39 contributed to the norovirus-inactivating effect. Notably, the hydrophobicity and positive charges in this region contributed to the norovirus-inactivating effect, as evidenced by the norovirus inactivation test using mutated residues 5–39. These findings are novel and highlight the possible application of heat-denatured lysozyme as a disinfectant against norovirus in a wide range of food processes.

## Introduction

Norovirus is a single-stranded RNA (+) virus belonging to the family *Caliciviridae*. It is transmitted orally by infected people or contaminated food. It causes severe vomiting, diarrhea, and fever 24–48 h after infection [1]. The main foods associated with norovirus gastroenteritis are oysters and other bivalves, although recently, numerous outbreaks of norovirus caused by unheated food products, such as salads and ready-to-eat food, have been reported [2, 3]. Every

Grant in-Aid for Scientific Research (B) Grant Number 17H03872. The funder, Kewpie Corporation, provided support in the form of salary for Masahiro Ichikawa but did not have any additional role in the study design, data collection and analysis, decision to publish, or preparation of the manuscript.

**Competing interests:** One of the authors, Masahiro Ichikawa, is an employee of a commercial company (Kewpie Corporation). The funder, Kewpie Corporation, provided support in the form of salary for Masahiro Ichikawa but did not have any additional role in the study design, data collection and analysis, decision to publish, or preparation of the manuscript. This does not alter the authors' adherence to all of the PLOS ONE policies on sharing data and materials.

year, norovirus is responsible for 64,000 episodes of diarrhea requiring hospitalization, and up to 200,000 deaths of children < 5 years of age in developing countries [4].

Although many norovirus-inactivating methods have been reported, including thermal treatment, ultraviolet irradiation, high hydrostatic pressure, hypochlorous acid, and the use of food-derived components [1, 5–7]; these methods are suboptimal as they affect the taste and color of food products. Consequently, the development of an anti-norovirus disinfectant agent is an important issue for food hygiene.

Lysozyme is a single-chain polypeptide consisting of 129 amino acids [8]. It catalyzes the hydrolysis of peptidoglycan of gram-positive bacteria. It is contained in secretions, such as tears and saliva, and egg white [8]. In addition, lysozyme is extracted on an industrial scale from chicken egg white and is widely used as a food additive or as a raw material for pharmaceuticals.

We have reported that thermally denatured lysozyme (DL) inactivates norovirus [9]. The particle size of murine norovirus strain 1 (MNV-1), a surrogate for norovirus, shows an average expansion of 16.37 nm after exposure to DL for 1 h, and the N-terminal region of lysozyme possibly contributes to the inactivating effect [9]. Furthermore, DL inactivates norovirus in several foods and is also effective against the hepatitis A virus [10–12]. However, the conditions for optimal DL norovirus-inactivating effects and the underlying mechanism remain unclear.

This study aimed to analyze the norovirus-inactivating conditions and mechanisms of DL. We evaluated the conditions under which DL is highly effective against norovirus, together with the changes in the gene expression of norovirus-infected host cells infected with DL-treated MNV-1. We also analyzed the involvement of specific lysozyme domains in the antiviral effect. The data suggest that residues 5–39 of lysozyme contribute to the antiviral effect of DL. These observations will inform the use of DL as an anti-norovirus disinfectant of foods.

## Materials and methods

### Virus and cells

MNV-1, kindly provided by Dr. Herbert W. Virgin from Washington University, was propagated in RAW264.7 macrophages (ATCC® TIB-71™) cultured at 37˚C under 5% $CO_2$ in Dulbecco's modified Eagles medium (DMEM, FUJIFILM Wako Pure Chemical Corporation, Osaka, Japan) containing 5% (v/v) fetal bovine serum (FBS), penicillin (100 U/mL), and streptomycin (100 μg/mL). After confirmation of the cytopathic effect during incubation at 37˚C under 5% $CO_2$, the cells were subjected to four cycles of freezing and thawing, and were then centrifuged at $8000 \times g$ for 20 min. The supernatant was used as an MNV-1 stock solution and stored at –80˚C until use.

### Plaque assay for MNV-1 infectivity determination

The infectivity of MNV-1 was determined by a plaque assay, as described previously [9]. Briefly, RAW264.7 cells were seeded into 6-well plates (Falcon BD, Franklin Lakes, NJ) at approximately 6 log cells/mL in DMEM containing 5% (v/v) FBS, penicillin (100 U/mL), and streptomycin (100 μg/mL). The plates were incubated at 37˚C under 5% $CO_2$ for 18 h. Then, 500 μL/well of samples prepared as described below sections were added to the plates, and the plates were shaken at 20 r/min for 1 h. The inoculated sample was removed, and the wells overlaid with 2 mL of 1.5% (v/v) SeaPlaque agarose (Lonza Japan, Tokyo, Japan) in DMEM containing 5% (v/v) FBS. The plates were incubated at 37˚C under 5% $CO_2$ for 48 h. Then, 2 mL of 0.03% (v/v) neutral red solution (Sigma-Aldrich Japan, Tokyo, Japan) was added, and the plates incubated at 37˚C under 5% $CO_2$ for 1 h. Thus, visualized plaques were then counted.

## Varying pH values of DL preparations and MNV-1 inactivation

DL was prepared as previously described [11]. Briefly, egg white lysozyme (FUJIFILM Wako Pure Chemical) was suspended in distilled water at a concentration (w/v) of 0.2%, 1.0%, and 2.0%; the initial pH (3.7) was adjusted to 4.5, 5.5, 6.5, 7.5, and 8.5 by the addition of 1 N NaOH. After passing through a 0.2-μm filter, the filtrate was heated in an oil bath at 100°C for 40 min and then cooled on ice.

For the virus inactivation experiment, 500 μL of MNV-1 suspension [approximately 6 log plaque-forming units (PFU)/mL] was mixed with 500 μL of DL preparation, for the final DL concentration of 0.1%, 0.5%, and 1.0%. The mixtures were allowed to stand for 1 min (for 1.0% DL final concentration) or 1 h (for 0.1% and 0.5% DL final concentrations). The samples were then diluted 10-fold in DMEM to stop the virus inactivation reaction and analyzed by the plaque assay, as described above.

## Characterization of DL prepared at different pH values

Hydrophobicity, thiol group content, and circular dichroism (CD) spectra of the different DL preparations were analyzed. Hydrophobicity was measured by using protein stability and aggregation assay kit (PSA200K, Profoldin, Hudson, MA), according to the manufacturer's instructions. Sample excitation was measured at 550 nm and emission at 610 nm, using a spectrofluorometer (SH-9000, Corona Electric Co., Ltd., Ibaraki, Japan). Thiol group content was determined in a reaction mixture containing 192 μL of deionized water, 20 mM Tris-HCl at pH 8.0, 2.8 μL of Ellman's reagent [prepared by dissolving 4 mg of 5,5′-dithiobis-(2-nitrobenzoic acid) powder in 1 mL of Tris buffer at pH 7.0], and 200 μL of 1.0% (w/v) DL. After incubation at room temperature for 1 h in the dark, sample absorbance was measured at 412 nm by using a microplate reader (SH-1000 Lab, Corona Electric Co., Ltd.). CD spectra were acquired using a circular dichroism dispersometer (J-720, JASCO Corporation, Tokyo, Japan). For the analysis, DL preparation was diluted to 100 μg/mL in distilled water, and the spectra acquired in the range of 190 to 250 nm, with a scanning speed of 100 nm/min, a bandwidth of 1 nm, and integration frequency of 10.

## Gene expression analysis of RAW264.7 macrophages infected with DL-treated MNV-1

RAW264.7 cell monolayers ($1 \times 10^7$ cells/mL) in T25 cell flasks (Thermo Fisher Scientific K. K., Tokyo, Japan) were inoculated as described below.

Live MNV-1 (6 log PFU/mL) was used at a multiplicity of infection (MOI) of 0.01. For some experiments, MNV-1 (6 log PFU/mL) was inactivated by heating at 100°C for 40 min in an oil bath. The heat-treated MNV-1 was then used at MOI of 0.01 (infectivity as determined before heat inactivation). For another set of experiments, 100 μL of MNV-1 (6 log PFU/mL) was mixed with 100 μL of 2.0% of DL (adjusted at pH 6.5 before thermally-denaturation) and allowed to stand at room temperature for 60 min. The mixture was used at MOI of 0.01 (infectivity as before DL treatment).

The inoculated RAW264.7 cell monolayers were incubated at 37°C under 5% $CO_2$ for 24 h. The cells were then detached by using a cell scraper (Thermo Fisher Scientific) and centrifuged at $300 \times g$ for 5 min. Mock infections were performed using the cell culture medium [DMEM containing 5% FBS, penicillin (100 U/mL), and streptomycin (100 μg/mL)].

Total RNA was extracted from inoculated RAW264.7 macrophages using the RNeasy Mini kit (Qiagen, Hilden, Germany), according to the manufacturer's instructions. The extracted RNA was reverse-transcribed using the PrimeScript RT reagent kit (Takara Bio, Shiga, Japan). Briefly, 8 μL of sample RNA was mixed with 2 μL of 5× PrimeScript buffer, 0.5 μL of Prime-Script RT Enzyme mix I, 25 pmol of oligo dT primer, and 50 pmol of random hexamer

primers. The reverse-transcription was performed using a thermal cycle GeneAmp PCR system 9700 (GE Healthcare Japan, Tokyo, Japan), at 37˚C for 15 min, followed by a denaturation step at 85˚C for 5 s. The resulting cDNA was amplified by quantitative polymerase chain reaction (qPCR) to quantitate the expression levels of interferon β (*Ifnb*) and interleukin 6 (*Il6*) genes [13]. For the qPCR reaction, 2 μL of cDNA was mixed with 12.5 μL of 2 × TB Green Premix Ex Taq II (Takara Bio), 8.5 μL of distilled water, and 0.4 μM of primers specific for the *Ifnb* or *Il6* genes, as previously described [13]. Amplification was performed using a QuantStudio 3 Realtime PCR system (Life Technologies Japan Ltd., Tokyo, Japan) with 95˚C of denaturation for 30 s, followed by 40 cycles of 95˚C for 5 s and 60˚C for 30 s. Fold-changes in mRNA abundance were calculated by the ΔΔCt method, as previously described [14].

## Verification of the inactivation domain in lysozyme

It was previously reported that lysozyme (PDB ID: 6BO2) residues 5–39 inactivate MNV-1 [9]. In the current study, three peptide variants (R1, R2, and R3) of the target sequence (Lzm 5–39), a peptide containing lysozyme residues 1–35 (Lzm 1–35), and one containing residues 88–125 (Lzm 88–125) were designed (Table 1). All the designed peptides were synthesized by Eurofin K. K. (Tokyo, Japan).

The synthesized peptides were dissolved in distilled water to the same molarity as 2.0% (w/v) lysozyme, 14,307 of molecular weight, and heated in an oil bath at 100˚C for 40 min, then immediately cooled in ice. Next, 120 μL of MNV-1 solution at 6 log pfu/mL was mixed with 120 μL of the prepared heat-denatured peptides, and allowed to stand for 60 min at room temperature. The samples were immediately diluted 10-fold in DMEM and analyzed by the plaque assay, as described above.

## Statistical analysis

Experiments were independently performed in triplicate. Values for the CD spectra are expressed as the mean, and other values are expressed as the mean ± standard deviation (SD). Significant differences were analyzed by Dunnett's test by using Microsoft Excel ver. 1908 (Microsoft Japan, Tokyo, Japan). The significance threshold was set at $p < 0.05$.

## Results

### Relationship between pH value of DL preparations and MNV-1 inactivation

Exposure of MNV-1 to 1.0% DL prepared at different pH values resulted in a decrease of MNV-1 infectivity. The inactivation efficiency of DL increased with the preparation pH, with

**Table 1. Peptides used in the current study[a].**

| Peptide | Amino acid sequence | No. residues (AA) | $M_w$ |
|---|---|---|---|
| Lzm 5–39 | RCELAAAMKRHGLDNYRGYSLGNWVCAAKFESNFN | 35 | 3993.47 |
| Lzm 5–39 R1 | RCELAAAMKR**E**GLDNYRGYSLGNWVCAA**E**FESNFN | 35 | 3986.36 |
| Lzm 5–39 R2 | RCELAAAMKRHGLDNYRGYSLGN**N**VCAAK**N**ESN**N**N | 35 | 3855.22 |
| Lzm 5–39 R3 | RC**N**LAAAMKRHGL**N**NYRGYSLGNWVCAAKF**N**SNFN | 35 | 3962.46 |
| Lzm 1–35 | KVFGRCELAAAMKRHGLDNYRGYSLGNWVCAAKFE | 35 | 3962.54 |
| Lzm 88–125 | ITASVNCAKKIVSDGNGMNAWVAWRNRCKGTDVQAWIR | 38 | 4220.82 |

[a] Mutated amino acids relative to the original sequence (Lzm 5–39) are shown in bold and underlined. Theoretical molecular weight is provided.

the most pronounced effects observed at pH $\geq$ 6.5. (Fig 1A). The infectivity of MNV-1 after exposure to DL preparations at pH 4.5–8.5 was significantly lower than that of the control ($p < 0.01$). Furthermore, upon exposure to DL preparations at pH 6.5–8.5, the infectivity of MNV-1 was below the detection limit (1.0 log PFU/mL). Similarly, when MNV-1 was exposed to 0.1% and 0.5% of DL preparations at different pH for 60 min, the infectivity of MNV-1 tended to decrease as the pH of DL preparations increased (Fig 1B).

## Characteristics of DL prepared at different pH values

The surface hydrophobicity of native lysozyme and DL prepared at different pH values was determined. The fluorescence of DL samples was significantly higher than that of native lysozyme at any pH ($p < 0.01$) (Fig 2A). In addition, the fluorescence intensity of DL preparations tended to increase as the pH increased, with DL preparations at pH 7.5 showing the maximum value. Furthermore, assessment of the free thiol group content, based on sample absorbance, revealed that the absorbance of DL preparations was significantly higher than that of native lysozyme at any pH value ($p < 0.01$) (Fig 2B). Sample absorbance tended to increase with an increasing pH of the DL preparation.

The CD spectra of native lysozyme and DL preparations were analyzed. Negative maxima at 208 nm and 222 nm, derived from the α-helix, were observed in the native lysozyme and DL preparations at pH 3.7–5.5 but not in DL preparations at pH 6.5–8.5 (Fig 2C). This indicated reduced α-helix content in DL preparations at pH 6.5–8.5.

## Gene expression analysis of RAW264.7 macrophages infected with DL-treated MNV-1

RAW264.7 macrophages were infected with MNV-1 under various conditions, and the expression of selected cytokine genes was analyzed by qPCR. When MNV-1 was used at MOI = 0.01, the expression of the *Ifnb* and *Il6* genes significantly increased as compared to mock-infected

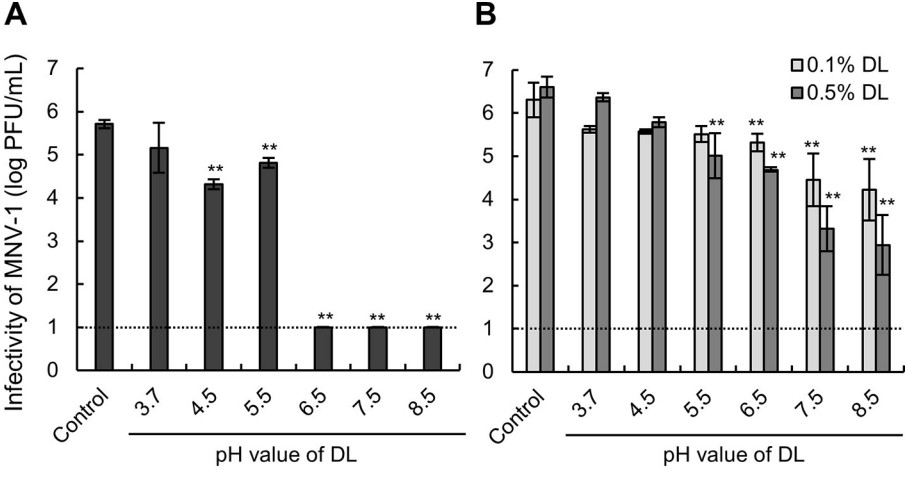

**Fig 1. Infectivity of MNV-1 after exposure to DL preparations at different pH values.** (A) MNV-1 was exposed to distilled water (Control) or 1% DL for 1 min. (B) MNV-1 was exposed to distilled water (Control), or 0.1%, or 0.5% of DL for 60 min. Values are expressed as the mean ± SD (*n* = 3). The dashed line indicates the detection limit of the plaque assay. Significant differences between the control and sample (exposed to DL) values were analyzed by Dunnett's test and shown; **$p < 0.01$.

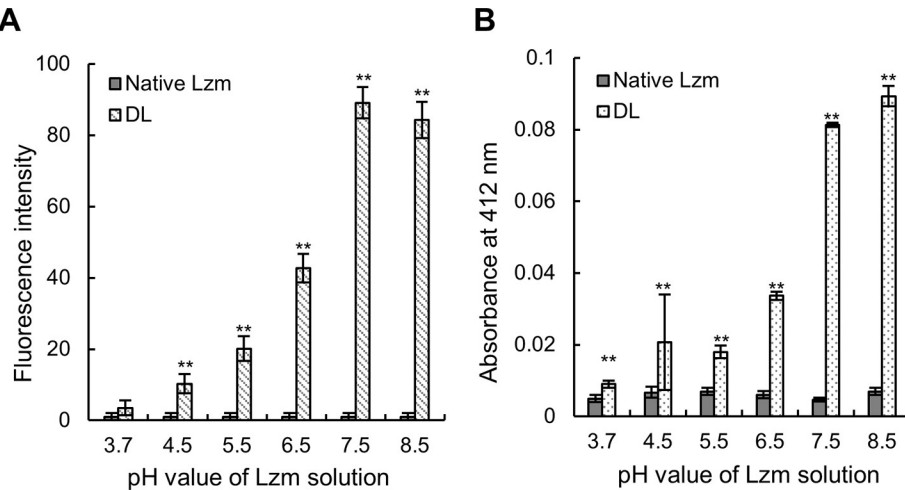

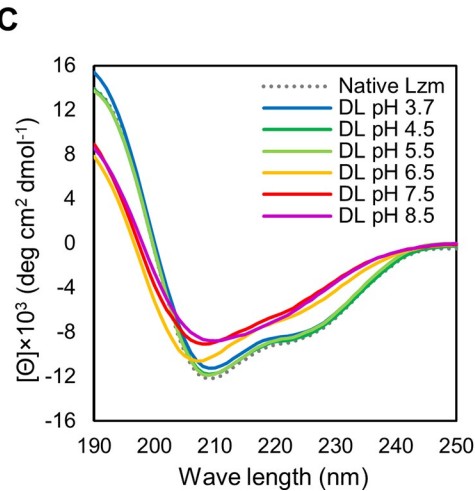

**Fig 2. Protein characteristics of DL at different pH values.** For the experiments, the pH of the native lysozyme (Native Lzm) solution was adjusted to pH 4.5–8.5 by using 1 N NaOH. DL (1%) was prepared by denaturation at 100°C for 40 min. (A) Surface hydrophobicity of 1% lysozyme solution prepared at different pH values. Samples were mixed with 1X PSA solution, and sample fluorescence intensity was determined ($\lambda_{Ex}$ = 550 nm, $\lambda_{Em}$ = 610 nm). Values are expressed as the mean ± SD ($n$ = 3). (B) Thiol group content in 1% lysozyme solution at different pH values. Samples were mixed with Ellman's reagent, and sample absorbance was determined at 412 nm. Values are expressed as the mean ± SD ($n$ = 3). (C) CD spectra of lysozyme preparations. The spectra are shown as an average of triplicate measurements. (A, B) Significant differences between native Lzm and DL preparations at the same pH value are shown; $^{**}p < 0.01$.

cells ($p < 0.01$, Fig 3). However, when MNV-1 had been inactivated by heat treatment or exposed to DL prior to the infection, the expression of the two genes did not increase (Fig 3).

## Verification of the lysozyme inactivation domain

Three variants of Lzm 5–39 were designed, and their inactivating effects against MNV-1 were evaluated. In the R1 variant, few basic amino acids were replaced with acidic amino acids to set the acidic/basic amino acid ratio to 1. As a result of amino acid substitution, the charge of the R1 variant decreased from +3 (Lzm 5–39) to -1. In addition, its hydropathy index [15] was -18.1, which was similar to that of Lzm 5–39 (-18.2).

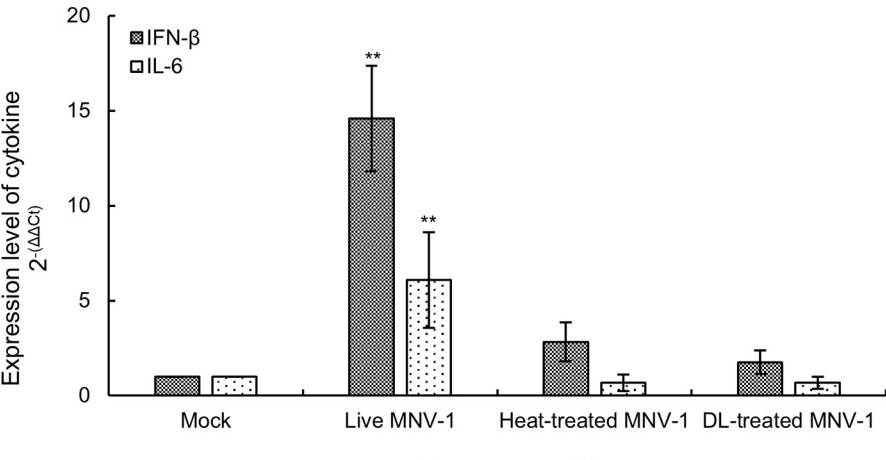

**Fig 3. Expression of the interferon-β and interleukin-6 genes in RAW264.7 macrophages infected with MNV-1.**
The macrophages were infected with MNV-1 at MOI = 0.01. The following MNV-1 preparations were used: live MNV-1; MNV-1 heat-inactivated at 100˚C for 10 min (Heat-treated MNV-1); and MNV-1 that had been exposed to 1.0% DL and adjusted to a pH of 6.5 before thermal denaturation for 60 min (DL-treated MNV-1). Mock treatment involved incubation with DMEM containing 5% FBS, penicillin (100 U/mL), and streptomycin (100 μg/mL). Gene expression was determined by qPCR. Fold-changes (relative to mock treatment samples) were calculated using the ΔΔCt method. Values are expressed as the mean ± SD ($n$ = 3). Significant differences between the mock and other samples are shown; $^{**}p < 0.01$.

In the R2 variant, hydrophobic amino acids were replaced to reduce the hydrophobicity without changing the charge. As Lzm 5–39 has many hydrophobic amino acids, the replacement of all these amino acids might prevent peptide synthesis. Thus, only aromatic amino acids were replaced. The charge of the resulting R2 peptide was +3, the same as Lzm 5–39, and the hydropathy index changed to -33.2.

In the R3 variant, all acidic amino acids were replaced with uncharged amino acids. The replacement resulted in a net charge of +6. Furthermore, two additional peptides (Lzm 1–35, and Lzm 88–125) from a different region of the lysozyme were synthesized, and the respective MNV-1 inactivating effects were also evaluated. The hydropathy plot was computed using the ProtScale tool with Kyte & Doolittle scale [15] in ExPASy [16] (S1 Fig).

Fig 4 shows a decrease in MNV-1 infectivity after exposure to DL (Lzm 1–129), Lzm 5–39, and replacement variants (R1–R3) after thermal denaturation. The infectivity of MNV-1 decreased by 0.9 log PFU/mL for Lzm 1–129 (pH 3.7, a final concentration of 1.0%) and by 1.7 log PFU/mL for the heat-denatured Lzm 5–39. In the case of the R1 and R2 variants, the inactivating effect of heat-denatured peptides against MNV-1 was lost; however, for the R3 variant, the MNV-1 infectivity was reduced by 2.0 log PFU/mL. The inactivating effect of the R3 variant was higher than that of Lzm 5–39 and the entire protein (Lzm 1–129). The MNV-1 infectivity was reduced by 1.7 and 0.2 log PFU/mL, by Lzm 1–35 and Lzm 88–125, respectively. Furthermore, both Lzm 1–35 and Lzm 5–39 treatments similarly reduced MNV-1 infectivity (Fig 4).

Moreover, CD spectrum analysis of Lzm 5–39 revealed that the negative maxima at 208 nm and 222 nm, derived from the α-helix and observed in the unheated sample, were reduced in the heat-denatured sample (S2 Fig).

## Discussion

In the present study, we aimed to analyze norovirus-inactivating conditions and the mechanism of DL. We showed that (1) the inactivating effect of DL was increased by adjusting the

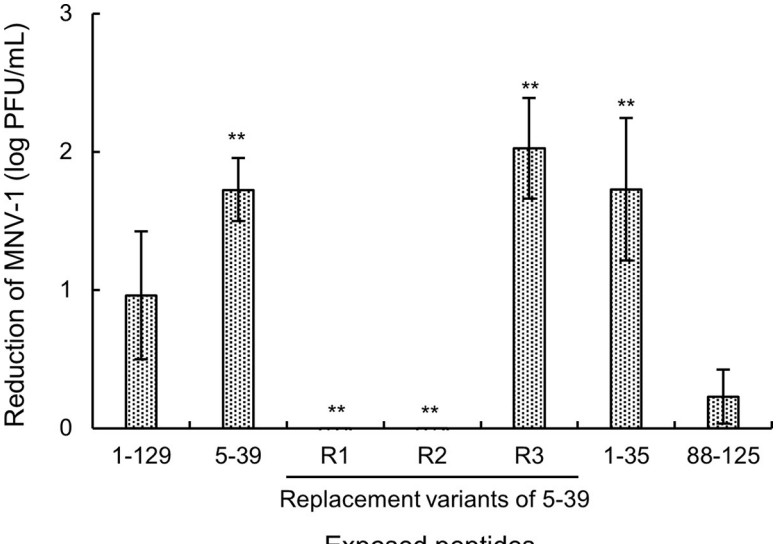

**Fig 4. MNV-1 inactivation by heat-denatured Lzm peptides.** Different lysozyme peptides were analyzed. All peptides and the entire lysozyme (amino acids 1–129) at the same molar concentration as that of 1.0% (w/v) lysozyme were heat-denatured at 100˚C for 40 min. MNV-1 (approximately 6 log PFU/mL) was mixed with DL (1%, w/v, pH 3.7) or the peptides for 60 min. The values are expressed as the mean of log reduction of MNV-1 ± SD ($n = 3$). Significant differences between the reduction in the infectivity of the whole protein (1–129) and that of each sample are shown; **$p < 0.01$.

pH to 6.5 or higher before thermal denaturation; (2) MNV-1 treated with DL did not affect the innate gene expression of host cells; and (3) the hydrophobicity of the protein structure. Of note, residues 5–39 primarily contributed to the antiviral effect of DL. These observations imply that the hydrophobic amino acids (particularly residues 5–39) of lysozyme, after exposure to thermal denaturation, interact with the structural protein of MNV-1, which contains multiple hydrophobic amino acids, thereby leading to the inactivation of MNV-1.

Although it was previously shown that the inactivating effect of DL is enhanced by increasing the heating temperature and heating time [9], other conditions may also contribute to the inactivating effect of DL. Therefore, to comprehensively evaluate the conditions under which DL is highly active against norovirus, the effect of the pH of the DL preparation was examined. As the pH of the lysozyme solution increased, the inactivating effect of DL also increased (Fig 1). The hydrophobicity and thiol group content significantly increased with the pH of the DL solution (Fig 2A and 2B).

Lysozyme is a positively charged protein with an isoelectric point of 11. We anticipated that in a high-pH solution (pH 6.5–8.5), the intermolecular forces would become stronger, thus, denaturing the protein structure upon heating. In other words, adjusting the pH to a high value would trigger the loosening of the protein structure during thermal denaturation. As shown in Fig 2A and 2B, the higher the pH, the higher the hydrophobicity and thiol group content of DL. This suggests that thermal denaturation disrupts the protein structure, exposing the otherwise buried hydrophobic amino acids. Of note, lysozyme also has many disulfide bonds, with multiple hydrophobic amino acids in the vicinity of cysteine residues [8].

We, therefore, speculate that the disulfide bonds were broken by thermal denaturation, exposing the nearby hydrophobic amino acids. As determined by the CD spectra of DL, heat denaturation reduced the α-helical content of the protein (Fig 2C), also indicating considerable changes in the secondary DL structure during thermal denaturation at pH 6.5 or higher.

We further evaluated the innate expression of host cells infected by DL-treated MNV-1. According to Enosi Tuipulotu et al. [13], MNV infection of RAW264.7 macrophages perturbs the transcriptional profile of host genes involved in interferon signaling, viral recognition, and cytokine stimulation, including the *Ifnb* and *Il6* genes. When RAW264.7 macrophages were infected with heat-inactivated MNV-1, MNV-1 was not recognized and did not affect the transcription of *Ifnb* and *Il6* genes (Fig 3). Similarly, DL-treated MNV-1 did not induce cytokine expression in macrophages. This indicated that the interaction between DL and MNV-1 was irreversible, and the treated virus was unable to enter and replicate in the host cell. This suggests that MNV-1 was completely inactivated by DL and did not regain infectivity.

Finally, to identify the virus-inactivating domain of lysozyme, we designed Lzm 5–39 replacement peptides, namely variants R1–R3 (Table 1). Data shown in Fig 4 suggest that the hydrophobicity and positive charge of Lzm 5–39 contribute to the inactivating effect of DL. An increase in the positive peptide charge could have enhanced the inactivating effect of the R3 variant, as it facilitated the absorption of the negatively charged virus particles by DL [17].

We also examined the effects of the Lzm 1–35 peptide in comparison with the Lzm 5–39 peptide. Lysozyme residues 1–4 form a random coil region containing three hydrophobic amino acids (VFG) (Table 1). Therefore, we expected that the virus-inactivating activity of Lzm 1–35 would be higher than that of Lzm 5–39. However, the activity was equivalent to that of Lzm 5–39 (Fig 4), which implied that the hydrophobicity of the heated protein structure was more important than the hydrophobicity of the primary structure. Based on the results shown in Fig 4, results in Fig 2A and 2B imply that the heat denaturation-related increase in surface-exposed hydrophobicity of DL contributes to the virus-inactivating effect, not merely hydrophobicity of DL.

We also examined the Lzm 88–125 peptide (encompassing three C-terminal helical structures of lysozyme). However, the peptide reduced MNV-1 infectivity by only 0.2 log PFU/mL, which was significantly lower than the effect observed for other peptides (Fig 4). This suggests that the structure of the N-terminal region of lysozyme contributes to viral inactivation, and that the hydrophobicity and positive charges of Lzm 5–39 region contribute to the inactivating effect of DL. This is supported by the observation that the α-helical content of heat-denatured Lzm 5–39 was lower than that of unheated Lzm 5–39 (S2 Fig).

Lysozyme lyses gram-positive bacteria by hydrolyzing the cell wall peptidoglycan, acting as a muramidase [18]. Ibrahim et al. [19] reported that heat-treated (80˚C for 30 min) lysozyme inactivates gram-negative bacteria. According to Sugahara et al. [20], the inactivating effect of DL against gram-negative bacteria is not associated with enzymatic activity but rather, with the hydrophobic amino acid residues of lysozyme exposed during heat denaturation and their association with the outer membrane of gram-negative bacteria. These considerations are in line with the data presented herein. An exposed loop region in the P domain corresponding to the surface of an MNV-1 structural protein contains multiple hydrophobic amino acids [21]. Therefore, we propose that the interaction of these hydrophobic amino acids with the hydrophobic amino acids of lysozyme exposed by thermal denaturation leads to MNV-1 inactivation.

Ibrahim et al. [22] reported that residues 46–61, 62–68, and 98–112 of lysozyme are deamidated by thermal denaturation and easily bind to monovalent and divalent cations. Therefore, these regions are considered to be the inactivating domain of DL effective against gram-negative bacteria. However, as shown in the current study, lysozyme residues 5–39 inactivated MNV-1, whereas residues 88–125 exerted no such inactivating effect (Fig 4). We, therefore, propose that although the mechanism of norovirus inactivation by lysozyme is similar to that of gram-negative bacteria inactivation, the lysozyme inactivation domain differs in the two cases.

Based on the presented data, (1) the MNV-1–inactivating effect of DL is enhanced by adjusting the pH of the lysozyme solution to 6.5 or higher before thermal denaturation; (2) the reaction of DL and MNV-1 is irreversible, and MNV-1 is completely inactivated by DL; and (3) the hydrophobicity and positive charge of lysozyme contribute to virus inactivation by DL. Hydrophobicity and positive charge are contradictory protein properties, and further detailed studies are required to resolve this observation. Hydrophobicity was implied to be more important in the protein structure than the hydrophobicity as the primary structure. Nonetheless, the findings of the current study may inform the practical use of DL as a disinfectant against norovirus. For instance, adjusting the pH conditions before thermal denaturation might aid the development of a cost-effective disinfectant, and a lysozyme fragment synthesized industrially might be useful as a novel disinfectant. We foretell that in the near future, DL might be used as a disinfectant in a wide range of food processes.

## Supporting information

**S1 Fig. Hydropathicity of synthesized peptides used in the current study.** Hydropathicity was computed by ExPASy [16] ProtScale using the Kyte & Doolittle scale [15]. (A) Lzm 5–39, (B) Lzm 5–39 variants (R1, R2, and R3), (C) Lzm 1–35, and (D) Lzm 88–125.
(TIF)

**S2 Fig. CD Spectra of a peptide containing lysozyme residues 5 to 39 (Lzm 5–39).** Peptide concentration was adjusted to 900 μM and heat-denatured at 100˚C for 40 min (Heated 5–39) or not (Unheated 5–39). The spectra are an average of triplicate measurements.
(TIF)

## Acknowledgments

The authors thank Akira Takeuchi, Moemi Nakazawa, Ryou Sasahara, and Hiroyuki Shidara from Kewpie Corporation for valuable advice and discussion on this research. We also thank Kanako Tsukidate, Natsuki Yamaguchi, and Yayoi Chujo for their assistance with the experiments.

## Author Contributions

**Conceptualization:** Michiko Takahashi, Hajime Takahashi.

**Data curation:** Michiko Takahashi.

**Formal analysis:** Michiko Takahashi.

**Funding acquisition:** Michiko Takahashi, Hajime Takahashi, Takashi Kuda, Bon Kimura.

**Investigation:** Michiko Takahashi, Yumiko Okakura, Masahiro Ichikawa.

**Methodology:** Michiko Takahashi, Yumiko Okakura.

**Project administration:** Michiko Takahashi, Hajime Takahashi, Bon Kimura.

**Resources:** Hajime Takahashi, Masahiro Ichikawa, Takashi Kuda, Bon Kimura.

**Supervision:** Hajime Takahashi, Takashi Kuda, Bon Kimura.

**Validation:** Michiko Takahashi.

**Visualization:** Michiko Takahashi.

**Writing – original draft:** Michiko Takahashi.

**Writing – review & editing:** Michiko Takahashi, Hajime Takahashi.

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
