## [Decision Letter · Decision Letter 0]

16 Jul 2020

PONE-D-20-14783

Impact of pH and protein hydrophobicity on norovirus inactivation by heat-denatured lysozyme

PLOS ONE

Dear Dr. Takahashi,

Thank you for submitting your manuscript to PLOS ONE. After careful consideration, we feel that it has merit but does not fully meet PLOS ONE’s publication criteria as it currently stands. Therefore, we invite you to submit a revised version of the manuscript that addresses the points raised during the review process.

Both reviewers are quite positive about the work but they both raised a number of issues that have to be addressed before the manuscript becomes acceptable.

We look forward to receiving your revised manuscript.

Kind regards,

Oscar Millet

Academic Editor

PLOS ONE

Journal Requirements:

"This work was supported by Grant-in-Aid for JSPS Fellows Grant Number 17J05482,

and Grant-in-Aid for Scientific Research (B) Grant Number 17H03872."

We note that one or more of the authors are employed by a commercial company: "Kewpie Corporation,"

Reviewers' comments:

Reviewer's Responses to Questions

**Comments to the Author**

1. Is the manuscript technically sound, and do the data support the conclusions?

Reviewer #1: Yes

Reviewer #2: Yes

2. Has the statistical analysis been performed appropriately and rigorously? 

Reviewer #1: Yes

Reviewer #2: Yes

3. Have the authors made all data underlying the findings in their manuscript fully available?

Reviewer #1: Yes

Reviewer #2: Yes

4. Is the manuscript presented in an intelligible fashion and written in standard English?

Reviewer #1: Yes

Reviewer #2: Yes

5. Review Comments to the Author

Reviewer #1: In previous work (10.1038/srep11819), the authors demonstrated that thermal/structural denaturation activates residues Lz23-57 (Lzm 5-39?) of egg white lysozyme in order to inactivate norovirus, a food contaminant. The present work explores the optimal inactivating conditions of denatured lysozyme. Optimal and complete inactivation of the glycoside hydrolase was found by raising the pH of the egg-white lysozyme solution above 6.5 prior to irreversible thermal denaturation, which reduced alpha helical content in CD spectra. Hydrophobic residues spanning the region 5-39 in the lysozyme sequence were posited to contribute to inactivation based on data for three mutant constructs. These findings are of relevance to the use of disinfectants in industrial food production.

1) It is not clear why the residues are numbered differently in the two reports by the authors (Lz 23-57 vs. Lzm 5-39).

2) The rationale for why/how the authors chose the three peptide variants is not entirely clear. The overall basic/positive peptide (6 basic, 3 acidic groups, pI = 11) was made to be either R1: neutral and amphiphilic (H->E and K->E mutations); R2: hydrophilic instead of hydrophobic (Asn instead of one Trp and two Phe); or R3: polar and more positive (3 acidic sidechains replaced with Asn). For instance, unlike Phe, tryptophan is a polar residue. Inactivation was lost with R1 and R2 and reduced in R3. It seems an oversimplification to say that the results show that 5-39 “hydrophobicity” is responsible for inactivation. Please describe the mutations/variants in the Abstract.

3) Line 310: it is presumably the “surface-exposed” hydrophobic and thiol content that increases with denaturation. The logic of mere “increased hydrophobicity” particularly breaks down in Lines 336-342. Lzm 1-35 and Lzm 5-39 were equivalent, so it seems that structural/surface changes upon denaturation are the culprit rather than the mere presence of hydrophobic residues (such as VFG). The authors propose that the exposed hydrophobic sequences bind the norovirus capsid rather than lysozyme having enzymatic activity, similar to the mechanism of gram-negative activation.

4) The readability of the manuscript could be improved by reorganizing passages. Methodological details are included in the Figure captions. The paragraph in the Discussion describes the effect of the mutant variants on protein charge, but this rationale is not explained in the earlier Introduction/Methods/Results sections. The Discussion would benefit from breaking up into subheadings/sections.

Reviewer #2: In this manuscript, Takahashi et al attempt to delineate the mechanism of inactivation of murine norovirus by heat-denatured hen egg white lysozyme (DL). They show that increasing the pH to 6.5-8.5 before thermal denaturation increase the effectiveness of DL. It is known from an earlier study that residues 5-39 was the most effective at virus inhibition. Here the authors introduce mutations to modify net charge and hydrophobicity and conclude that both hydrophobicity as well positive charge of this peptide contribute to viral inhibition. It is also interesting that a simple clustering of hydrophobic residues in the primary sequence is not sufficient, rather the results with peptide 88-125 in this paper along with alpha-lactalbumin from a previous study (Takahashi et al, Scientific Reports, 2015) suggest that exposure of hydrophobic and charged residues in a particular conformation must be responsible for this. Of course, it is hard to imagine a ‘conformation’ in a heat denatured protein, and hence further studies are needed.

Overall, the results presented in the paper are interesting, clear and understandable, and backed with evidence. As such it merits publication in Plos One, however a few issues need to be addressed before acceptance.

1. It would be good to include a hydropathy plot for each peptide along with the mutant variants.

2. Does the 5-39 peptide become totally unstructured in the presence of some denaturant? It is easy to get that information from a CD spectrum in the presence of some urea. This experiment will confirm that even the heat denatured peptide has some residual structure. Also, can you obtain CD spectrum of the heat denatured peptide in multiple batches and overlay? This will also make sure that this residual conformation is consistent every time you denature different batches.

3. You mention that increase in thiol content upon denaturation might be due to SS bonds breaking during heating. While this is possible, it would be great to quantify how many actually break (using Ellman’s reagent). This should be straight forward to do.

4. The organization of the manuscript needs to be changed. For example, there are several details about peptide design, and interpretation which are in the Methods and Discussion, while ideally, they should all be part of the Results. It is very difficult to go back and forth while reading. Methods should not contain the logic behind doing an experiment, rather should only focus on the technical details. Results of hydropathy index calculation are all included in Discussion, which again should move to Results.

5. The word ‘Denatured’ in keywords is misspelt.

6. PLOS authors have the option to publish the peer review history of their article (what does this mean?). If published, this will include your full peer review and any attached files.

Reviewer #1: **Yes: **Ron Hills

Reviewer #2: No

---

## [Author Response · Author response to Decision Letter 0]

31 Jul 2020

Reponses to reviewer #1

1) It is not clear why the residues are numbered differently in the two reports by the authors (Lz 23-57 vs. Lzm 5-39).

Thank you for your comments and suggestions. We found them quite useful as we approached our revision.

- Sorry for confusing. In the previous report (Takahashi et al., 2015. Sci. Rep. 5: 11819), the reference sequence of lysozyme (GenBank ID: AAL69327) is a chicken lysozyme chromatin domain containing a lysozyme sequence at residues 19-145 (Chong et al., 2002. Nucleic Acids Res. 30:463-467). In this sequence, residues of 23-57 correspond to Lzm 5-39 of the lysozyme sequence (PDB ID: 6BO2). 

2) The rationale for why/how the authors chose the three peptide variants is not entirely clear. The overall basic/positive peptide (6 basic, 3 acidic groups, pI = 11) was made to be either R1: neutral and amphiphilic (H->E and K->E mutations); R2: hydrophilic instead of hydrophobic (Asn instead of one Trp and two Phe); or R3: polar and more positive (3 acidic sidechains replaced with Asn). For instance, unlike Phe, tryptophan is a polar residue. Inactivation was lost with R1 and R2 and reduced in R3. It seems an oversimplification to say that the results show that 5-39 “hydrophobicity” is responsible for inactivation. Please describe the mutations/variants in the Abstract.

- As you pointed out, the description about the background of peptide-design was not sufficient in the manuscript.

- In the R1 variant, in order to make the number of acidic and basic amino acids the same, 2 out of 6 basic amino acids were replaced with acidic amino acids. Here, the hydropathy index was calculated to confirm that the hydropathy was almost the same as before replacement.

- The R2 variant was aimed to change only the hydrophobicity without changing the charge. Lzm 5-39 originally had a large number of hydrophobic amino acids, and the replacement of all of the residues might prevent peptide-synthesis. Therefore, we decided to replace only the aromatic amino acids, which has a particularly high degree of hydrophobicity.

- In the R3 variant, all the acidic amino acids were replaced with uncharged amino acids in order to shift the basic side.

- The background of peptide design was added into the revised manuscript (P. 15, L. 253-268 in Revised Manuscript with Track Change). We also revised abstract section to describe the peptide mutation/variants (P. 2, L. 31-37 in Revised Manuscript with Track Change).

- As you commented, it cannot be concluded that only hydrophobicity contributes to virus-inactivating effect from the Fig. 4. Further study is needed to elucidate which of the characters, positive charge and hydrophobicity, dominates the virus-inactivation mechanism. It has been described in the manuscript (P. 22, L. 379-382 in Revised Manuscript with Track Change). 

3) Line 310: it is presumably the “surface-exposed” hydrophobic and thiol content that increases with denaturation. The logic of mere “increased hydrophobicity” particularly breaks down in Lines 336-342. Lzm 1-35 and Lzm 5-39 were equivalent, so it seems that structural/surface changes upon denaturation are the culprit rather than the mere presence of hydrophobic residues (such as VFG). The authors propose that the exposed hydrophobic sequences bind the norovirus capsid rather than lysozyme having enzymatic activity, similar to the mechanism of gram-negative activation.

- Fig. 4 implies that the surface-exposed hydrophobicity is important for virus inactivation, rather than the mere hydrophobicity of heat-denatured lysozyme as your comment. On the other hand, since it could not be asserted that “surface-exposed hydrophobicity is important” with Fig. 2 only. In response to the suggestion, we added sentence “” after the consideration of Fig. 4 (P. 20 L. 345-347 in Revised Manuscript with Track Change). 

4) The readability of the manuscript could be improved by reorganizing passages. Methodological details are included in the Figure captions. The paragraph in the Discussion describes the effect of the mutant variants on protein charge, but this rationale is not explained in the earlier Introduction/Methods/Results sections. The Discussion would benefit from breaking up into subheadings/sections.

- The structure of paragraph was reorganized as suggested by both reviewer #1 and reviewer #2. The description of mutant peptides, hydropathy index calculation, and hydropathy plot have been moved to the results section based on the proposal of reviewer #2 (P. 15 L. 253-268 in Revised Manuscript with Track Change). 

 

Responses to reviewer #2

1. It would be good to include a hydropathy plot for each peptide along with the mutant variants.

We appreciate your comments and suggestions, which have helped us significantly improve the paper. We tried to be responsive to your concerns.

- Hydropathy plot for each peptide was added in this revision as suggested (S Fig. 1). 

2. Does the 5-39 peptide become totally unstructured in the presence of some denaturant? It is easy to get that information from a CD spectrum in the presence of some urea. This experiment will confirm that even the heat denatured peptide has some residual structure. Also, can you obtain CD spectrum of the heat denatured peptide in multiple batches and overlay? This will also make sure that this residual conformation is consistent every time you denature different batches.

3. You mention that increase in thiol content upon denaturation might be due to SS bonds breaking during heating. While this is possible, it would be great to quantify how many actually break (using Ellman’s reagent). This should be straight forward to do.

- As your suggestion, acquiring CD spectrum in the presence of some urea and quantifying SS bonds could further enhance the manuscript. However, unfortunately we are unable to perform these additional experiments due to the current COVID-19 pandemic. 

4. The organization of the manuscript needs to be changed. For example, there are several details about peptide design, and interpretation which are in the Methods and Discussion, while ideally, they should all be part of the Results. It is very difficult to go back and forth while reading. Methods should not contain the logic behind doing an experiment, rather should only focus on the technical details. Results of hydropathy index calculation are all included in Discussion, which again should move to Results.

- We reorganized manuscript as your suggestion. In particular, some descriptions about peptide design, hydropathy index calculation, and hydropathy plot were moved to in the results section (P. 15, L. 253-268 in the Revised Manuscript with Track Change). 

5. The word ‘Denatured’ in keywords is misspelt.

- “Denatured” in keywords was corrected in this revision.

Again, thank you for taking the time and energy to help us improve the paper. We are hopeful that our revision helps to improve your opinion of work.

---

## [Editor Report · Decision Letter 1]

5 Aug 2020

Impact of pH and protein hydrophobicity on norovirus inactivation by heat-denatured lysozyme

PONE-D-20-14783R1

Dear Dr. Takahashi,

We’re pleased to inform you that your manuscript has been judged scientifically suitable for publication and will be formally accepted for publication once it meets all outstanding technical requirements.

Kind regards,

Oscar Millet

Academic Editor

PLOS ONE
---

## [Editor Report · Acceptance letter]

10 Aug 2020

PONE-D-20-14783R1 

Impact of pH and protein hydrophobicity on norovirus inactivation by heat-denatured lysozyme 

Dear Dr. Takahashi:

I'm pleased to inform you that your manuscript has been deemed suitable for publication in PLOS ONE. Congratulations! Your manuscript is now with our production department. 

Kind regards, 

on behalf of

Dr. Oscar Millet 

Academic Editor

PLOS ONE